# Restriction of Zika Virus Replication in Human Monocyte-Derived Macrophages by Pro-Inflammatory (M1) Polarization

**DOI:** 10.3390/ijms26030951

**Published:** 2025-01-23

**Authors:** Isabel Pagani, Silvia Ghezzi, Giulia Aimola, Paola Podini, Francesca Genova, Elisa Vicenzi, Guido Poli

**Affiliations:** 1Viral Pathogenesis and Biosafety Unit, Division of Immunology, Transplantation and Infectious Diseases, IRCCS San Raffaele Scientific Institute, Via Olgettina 58, 20132 Milan, Italy; pagani.isabel@hsr.it (I.P.); ghezzi.silvia@hsr.it (S.G.); vicenzi.elisa@hsr.it (E.V.); 2Human Immuno-Virology (H.I.V.) Unit, Division of Immunology, Transplantation and Infectious Diseases, IRCCS San Raffaele Scientific Institute, Via Olgettina 58, 20132 Milan, Italy; giulia.aimola@gmail.com; 3Division of Neuroscience, Institute of Experimental Neurology, IRCCS San Raffaele Scientific Institute, Via Olgettina 58, 20132 Milan, Italy; podini.paola@hsr.it; 4Center for Omics Sciences, IRCCS San Raffaele Scientific Institute, Via Olgettina 58, 20132 Milan, Italy; genova.francesca@hsr.it; 5School of Medicine, Vita-Salute San Raffaele University, Via Olgettina 58, 20132 Milan, Italy

**Keywords:** Zika virus, monocytes, macrophage polarization, viral restriction, receptors, interferon-stimulated genes

## Abstract

Zika virus (ZIKV), a member of the Flaviviridae family, is primarily transmitted through mosquito bites, but can also spread via sexual contact and from mother to fetus. While often asymptomatic, ZIKV can lead to severe neurological conditions, including microcephaly in fetuses and Guillain–Barré Syndrome in adults. ZIKV can infect placental macrophages and fetal microglia in vivo as well as human monocytes and monocyte-derived macrophages (MDMs) in vitro. Here, we observed that both human monocytes, and MDM particularly, supported ZIKV replication without evident cytopathicity, with virions accumulating in cytoplasmic vacuoles. We also investigated whether the cytokine-induced polarization of MDMs into M1 or M2 cells affected ZIKV replication. The stimulation of MDMs with pro-inflammatory cytokines (interferon-γ and tumor necrosis factor-α) polarized MDMs into M1 cells, significantly reducing ZIKV replication, akin to previous observations with a human immunodeficiency virus type-1 infection. In contrast, M2 polarization, induced by interleukin-4, did not affect ZIKV replication in MDMs. M1 polarization selectively reduced the expression of MERTK, a TAM family putative entry receptor, and increased the expression of several interferon-stimulated genes (ISGs) previously associated with the containment of ZIKV infection; of interest, ZIKV infection transiently boosted the expression of some ISGs in M1-MDMs. These findings suggest a dual mechanism of ZIKV restriction in M1-MDMs and highlight potential antiviral strategies targeting innate immune responses.

## 1. Introduction

Zika virus (ZIKV), a flavivirus, was firstly isolated from a febrile sentinel rhesus monkey in Uganda in 1947 [1] and it has generated episodic global health concerns over time [2]. Attention was drawn to it during the 2016 Brazilian outbreak of microcephaly—a severe congenital anomaly—in newborns when a direct correlation with ZIKV infection in pregnant women was established, underscoring its potential for causing widespread health crises [3,4]. Despite reduced reported cases post-2016, ZIKV persists in over 80 countries, predominantly in equatorial regions favorable to its primary vectors, i.e., *Aedes* mosquitoes (https://cdn.who.int/media/docs/default-source/documents/emergencies/zika/zika-epidemiology-update_february-2022_clean-version.pdf?sfvrsn=c4cec7b7_1&download=true, accessed on 20 January 2025). This persistence of ZIKV underscores its enduring public health threat, complicated by its multiple infection modes, including the dominant vector-borne [5], but also sexual and vertical transmissions [6]. The latter is particularly concerning due to its association with Congenital Zika Syndrome (CZS), a spectrum of neurological disorders in infants resulting from in utero exposure to ZIKV [7].

Mononuclear phagocytes (MPs), encompassing monocytes and macrophages, are integral to the body’s first line of defense against infectious agents, including several viral infections. Tissue-resident macrophages (TRMs) are credited with a dual origin; while most TRMs arise from embryonal structures, such as the yolk sac and fetal liver, circulating monocytes (as well as some TRMs) originate from the bone marrow [8]. In homeostatic conditions, TRMs are capable of self-renewal, thereby persisting through the lifetime of the host, whereas monocytes are short-lived cells that are eliminated by apoptosis without leaving the circulation [9]. However, upon infection or tissue damage, circulating monocytes are recruited in the tissue by chemotactic signals and rapidly differentiate into monocyte-derived macrophages (MDMs), which collaborate with TRMs to eliminate the source of disturbance [10]. MDMs are not credited with a self-renewing capacity like TRMs; therefore, upon the elimination of the pathogenic component (either infectious or not infectious) the tissue returns to its homeostatic condition with TRMs as solo sentinels. A second peculiar property of MPs is their capacity to activate differential activation/polarization programs as a function of the microenvironmental stimuli. This has been highlighted by the description of two primary phenotypes: M1, or “classically activated” cells, induced by pro-inflammatory cytokines and related signals, which play a crucial role in anti-viral defenses, and M2, or “alternatively activated” cells, characterized by an anti-inflammatory profile and the capacity to promote tissue repair and remodeling [11]. Although M1 and M2 are considered the two extremes of several intermediate functional states of macrophages [12], their comparison can inform on the plasticity of MPs in response to different insults. In this regard, we have previously characterized the differential capacity of M1- vs. M2-polarized human MDMs of containing the replication of the human immunodeficiency virus type-1 (HIV-1) and provided evidence that a second polarization of M1-MDMs with pro-inflammatory cytokines after HIV-1 infection drives them to a state of reversible latency [13,14]. Macrophage polarization is also believed to play a crucial role in pregnancy for maintaining maternal–fetal tolerance and supporting gestational progression [15]. In particular, macrophages in early pregnancy typically exhibit an anti-inflammatory, M2-like phenotype, which promotes tissue remodeling, placental development, and immune tolerance by suppressing inflammatory responses against the fetus, thereby preventing its rejection. As pregnancy advances, a shift toward a more pro-inflammatory, M1-like phenotype occurs aiding in labor initiation and parturition, thereby facilitating maternal health and successful fetal development [15].

While the susceptibility of human macrophages isolated from different tissue and organs, such as Hofbauer cells from the placenta [16,17], fetal brain microglia [18,19,20], and lymph node-resident macrophages [21], to ZIKV infection has been well established, it remains unclear whether the functional polarization of human MDMs into M1 or M2 phenotypes influences viral infection and replication.

To address this gap, our study aimed at investigating ZIKV infection of primary human monocytes and MDMs, focusing on whether the polarization of the latter affects virus replication.

## 2. Results

### 2.1. ZIKV Productively Infects Human Monocytes and, More Efficiently, MDMs

We firstly investigated the capacity of ZIKV to infect freshly isolated human monocytes and 7-days-old MDMs, as both cell types can potentially serve as viral reservoirs [22]. We tested in parallel two ZIKV strains: the old MR766 of African origin and the more recent PRVABC59, an Asian isolate; cells were incubated with infectious culture supernatants at the multiplicity of infection (MOI) of 1. Monocyte culture supernatants were collected 1 h after virus incubation and the residual viral input was estimated to be approximately 3 log_10_ for both viral strains, as determined by a plaque-forming assay (PFA); a decrease of approximately 1 log_10_, returning to the initial level at day 7 post-infection, was then observed (Figure 1a). When the cell cultures were terminated 9 days after infection, an additional log_10_ increase in infectious virus vs. the initial viral input was noted, thus indicating productive ZIKV replication (Figure 1a). The supernatant content in adenylate kinase (AK), a marker of necrotic cell death, in infected cell cultures was similar to that of control, uninfected monocyte cultures, thus indicating a lack of gross cytopathicity in association with virus replication (Figure 1b).

When MDMs were incubated with both viral isolates (MOI = 1), a productive infection was promptly observed without exhibiting the transient decrease in infectious virus release observed in monocyte cultures (Figure 1c). A net gain of 3 log_10_ of infectious virus above the initial input levels was observed up until day 9 post-infection when the cell cultures were terminated (Figure 1c). As for monocytes, no evidence of necrotic cell death was observed by the AK release assay (Figure 1d), despite the more robust levels of virus replication. No significant differences were noted in the kinetics of infection between the two ZIKV strains, whereas the superior capacity of MDMs to support ZIKV replication compared to monocytes was confirmed using cell cultures established from the same donors (Appendix A).

In summary, freshly isolated monocytes are susceptible to ZIKV infection and support virus replication, although with a lower efficiency compared to MDMs. In contrast to neural progenitor cells in which the high levels of ZIKV replication causes profound cytopathic effects [23], monocytes and MDMs support virus production, even at high levels in the case of MDMs, without evident cell damage.

### 2.2. Functional Polarization of Human MDMs into M1, but Not M2, Cells Restricts ZIKV Replication

The stimulation of monocytes or MDMs with pro-inflammatory cytokines, i.e., tumor necrosis factor-α (TNF α) plus interferon-γ (IFN-γ) or interleukin-4 (IL-4), for a short period of time (18 h) results in their polarization into M1 and M2 cells, respectively [13]. Previously, when M1- or M2-MDMs, or M1/M2 polarized monocytes [24], were infected with HIV-1, a restriction of virus replication was observed in comparison to unpolarized cells, with M1 causing a more profound, although less durable inhibition than M2 stimulation [13].

In this study, we differentiated human MDMs into M1 and M2 cells and subsequently infected them with either the MR766 or PRVABC59 strains (MOI = 1) after removing cytokine-containing supernatants. Cell culture supernatants were collected daily up to day 9 post-infection. While both control (CTRL), unpolarized, and M2-MDMs efficiently supported ZIKV replication (with no significant differences observed between these two conditions), no evidence of productive infection was observed in M1-MDMs. Specifically, the levels of infectious virus released in culture supernatants remained constant and even decreased by 1 log_10_ over the 9-day culture period (Figure 2a). These findings were further confirmed using an indirect immunofluorescence assay (IFA) with a monoclonal antibody (mAb) specific for the newly synthesized double-stranded viral RNA (Figure 2b). Quantification of IFA-positive cells revealed a significant reduction in M1-MDMs compared to M2-MDMs or CTRL MDMs (Figure 2c).

As observed in infected and unpolarized MDMs, no cytopathicity was detected in either infected or uninfected cells, regardless of their polarization status, as determined by the AK release assay (Figure 2d).

Thus, in partial analogy to what we observed with HIV-1 infection, the M1 polarization of human MDMs significantly impaired ZIKV replication, whereas M2 polarization had no effect.

### 2.3. M1 Polarization Induces Retention of ZIKV Particles in the Cytoplasm of Infected Cells

To elucidate the mechanism by which M1 polarization restricts ZIKV replication, we investigated both uninfected and ZIKV-infected MDMs by transmission electron microscopy (TEM) three days post-infection. Ultrastructural analysis revealed free virions in both CTRL-MDMs and M1 cells, albeit with a much higher abundance in the formers than in the latter (Figure 3a). Virions were engulfed in intracytoplasmic vacuoles. Of note, the presence of intracellular vacuoles containing virions has already been independently reported for the infection of Dengue Virus 2 (DENV-2), another flavivirus, in cultured human monocytes [25]. In uninfected control MDMs, empty vacuoles were observed (Figure 3a and Appendix A).

To quantify the levels of cell-associated virions, we subjected the infected cells to five sequential freeze–thaw (F/T) cycles to disrupt cell membranes and release entrapped virions, as previously reported for HIV-1 [26,27]. Viral RNA levels were then quantified using real-time PCR targeting the genomic RNA. Three days after infection, a substantial fraction of viral RNA (ca. 66% of the total) remained associated with the cells, rather than being released, in unpolarized MDMs. In contrast, M1 cells exhibited similar levels of cell-associated viral RNA and viral RNA in the supernatants (Figure 3b). Overall, these findings indicate that M1 polarization significantly reduces ZIKV RNA levels, both in terms of virions released into the culture supernatants and of those retained intracellularly.

### 2.4. M1 Polarization of MDMs Induces a Downregulation of MERTK, a Putative ZIKV Entry Receptor

While the exact mechanisms of ZIKV entry into target cells remain to be fully elucidated, members of the “TAM receptor family” (namely, TYRO3, AXL, and MERTK) have been identified as significant contributors, with particular emphasis on AXL [28,29,30]. Thus, we determined if one or more TAM receptors was present on the MDM’s surface and whether M1 or M2 cell polarization modulated their levels of expression. AXL was barely detectable, as determined by Western blot analysis (Figure 4a), whereas the proportion of AXL^+^ cells in both unpolarized and M1 or M2 MDMs did not exceed 2%, as quantified by flow cytometry (Figure 4b). TYRO3 was expressed in ca. 30% of control MDMs and was neither modulated in M1 nor in M2 cells, as determined by flow cytometry (Figure 4b). In contrast, MERTK was abundantly expressed in unpolarized MDMs and was either strongly or moderately downregulated in M1- and M2-MDMs, respectively (Figure 4a,b and Appendix A).

Thus, the downregulation of MERTK could potentially contribute to the overall restriction of ZIKV replication observed in infected M1-MDMs.

### 2.5. Increased Expression of Several Interferon Stimulated Genes (ISGs) in M1- vs. Control MDMs

As IFN-γ is a primary driver of the M1 polarization of MDMs [31], we investigated the expression of specific ISGs known to impair the replication of different viruses by RNA sequencing (RNAseq) [32]. M1-polarized MDMs indeed showed an upregulation of several ISGs previously associated with the inhibition of ZIKV infection and/or spreading [33,34,35,36,37], as shown in Figure 5a. When examining the expression of these ISGs at different time points post-infection in both CTRL and M1-MDMs, we observed a transient upregulation of these genes upon infection at 24 h post-infection that was particularly significant for ISG20 and OAS2 (Figure 5b), which was not sustained at 72 h post-infection.

Thus, the observation that the M1 polarization of MDMs was associated with the upregulation of several ISGs provides a second potential mechanism of restriction of ZIKV replication in human macrophages additional to the downregulation of MERTK-mediated cell entry.

## 3. Discussion

We here confirmed that human monocytes and, in particular, MDMs supported productive ZIKV infection in vitro in the absence of evident cytopathicity. MDM infection was associated with retention of ZIKV virions within the cytoplasm and in cytoplasmic vacuoles that accounted for up to 66% of total virus production. In addition, we observed that the M1 polarization of MDMs, induced by short-term stimulation with IFN-γ and TNF-α, significantly restricted ZIKV replication, whereas M2 polarization, induced by cell stimulation with IL-4, did not significantly influence virus production in comparison to control, unpolarized MDMs. M1 restriction was associated with a profound downregulation of MERTK, a putative entry receptor of the TAM family, whereas neither AXL nor TYRO3 showed evidence of modulation in polarized cells. Finally, M1 restriction was associated with the upregulation of several ISGs that could potentially contribute to the observed inhibition of ZIKV replication in MDMs.

Previous studies have demonstrated the ability of ZIKV to infect and replicate in human macrophages [16,17,38], suggesting that these cells could serve as sources of virus production, thereby facilitating the spread of the virus to various organs and tissues, including the placenta in cases of congenital transmission [39,40,41,42]. Similarly, monocytes have been shown to be susceptible to ZIKV infection, albeit with variable efficiency and outcomes [43,44,45]. In this regard, ZIKV infection of circulating monocytes is believed to play a crucial role in viral dissemination and pathogenesis, as these cells can transport the virus to various tissues, including the placenta and the developing central nervous system (CNS) of the fetus, thereby facilitating vertical transmission of ZIKV from the infected mother [46]. Furthermore, ZIKV infection can alter monocyte differentiation and function, shifting the balance between monocyte subsets, which may have implications for immune evasion and inflammatory responses [45].

In our studies, MDMs were considerably more permissive to ZIKV replication than freshly isolate monocytes suggesting that the differentiation into MDMs is required to support robust viral replication, as previously observed in HIV-1 infection [13,14]. In addition, similar to HIV-1, neither monocyte nor MDM infection was associated with overt cytopathicity, at least in terms of EM morphology and cell necrosis, as revealed by the AK release assay. This observation suggests that ZIKV can establish a replicative niche within macrophages across different tissues without significantly compromising cell viability, therefore favoring ZIKV persistence and dissemination within the host.

Macrophages, including circulating monocytes and MDMs, exhibit a sophisticated response to pathogens, activating different programs depending on environmental stimuli. This adaptability is exemplified by M1/M2 polarization analogous to the Th1/Th2 polarization previously defined for CD4^+^ T lymphocytes [47]. While M1 cells, also known as “classically activated”, are induced by pro-inflammatory signals and contribute to anti-viral responses, “alternatively activated” M2 cells are induced by several molecules, including IL-4, and promote tissue repair and regeneration in addition to quenching inflammation [31,48]. Currently, M1/M2 polarized macrophages are considered the two extremes of several intermediates phenotypes that human macrophages can undergo in response to environmental signals [49]. Our study has demonstrated that M1-polarized MDMs restrict ZIKV replication whereas M2-polarization did not influence virus production in these cells. This is in part analogous to what we have previously observed with HIV-1 infection in which both M1 and M2 polarization led to a restriction of virus replication [13], although accomplished by different modalities and with M1 cells exhibiting a more profound inhibition of different steps of the virus life cycle [50]. M1-MDMs showed a downregulation of MERTK, a putative entry receptor for ZIKV. In contrast, neither AXL nor TYRO3 showed evidence of a response to cell stimulation with the pro-inflammatory cytokines. Notably, flavivirus infections per se, including Japanese encephalitis virus, DENV, and ZIKV, can promote a type 1 macrophage (M1) polarization of brain macrophages, which is associated with the severity of encephalitis [51]. In the same study, the pharmacological blockade of TNF-α was shown to partly retard DENV-induced M1 polarization, suggesting a potential therapeutic strategy for mitigating the severity of flavivirus-induced encephalitis [51].

We observed the retention of ZIKV virions in the cytoplasm of infected MDMs accounting for ca. 66% of the total virus production in these cells. Moreover, membrane-coated cytoplasmic vacuoles containing viral particles resembling features already reported independently in DENV-2-infected monocytes were noted [25]. This feature was maintained in M1-MDMs, although the amounts of viral particles was much lower than that of unpolarized cells. It is important to underscore that this analysis does not allow the distinction between virions associated with the plasma membrane and cytoplasm-associated free virions. Furthermore, this approach is not informative on the proportion of infectious vs. defective virions. In this regard, we attempted to evaluate the infectivity of the cell-associated virions by the PFA, but the disruption of cell integrity by F/T cycles also abolished the infectivity of cell-associated virus.

Despite ZIKV’s ability to counteract the IFN response, which is crucial for establishing infection [52], the virus remains sensitive to the antiviral actions of many ISGs [53]. IFITM1 and IFITM3 were reported to interfere with ZIKV replication, likely acting in an early phase of the viral life cycle [54], whereas MXA overexpression exerted its inhibitory effects independently of viral entry and involving the induction of type 1 IFN-dependent signaling [35]. ISG15 has been shown to inhibit the replication of different flaviviruses, including ZIKV, and its effect was correlated to the stabilization of its binding partner USP18 that competes with the NS5-dependent degradation of STAT2 [36]. Noteworthy, two ISGs were further upregulated 24 h post-infection in M1-MDMs: ISG20 and OAS2. ISG20 exerts 3′-5′ exonuclease activity against different viruses, including flaviviruses, and it was rapidly induced in trophoblast cells after infection with ZIKV [37]. Furthermore, upon IFN stimulation, ISG20 and has been reported to promote additional antiviral effects against influenza virus [55], potentially through the interaction with other ISGs. OAS2 is an IFN-inducible enzyme involved in the detection of viral RNA and the subsequent activation of RNase L, which degrades both host cell and viral RNAs [56]. Several studies have underscored the importance of OAS2 in controlling viral replication, including ZIKV infection [34,57].

## 4. Materials and Methods

### 4.1. Isolation of Monocytes and Differentiation into MDMs

Peripheral blood mononuclear cells (PBMCs) were isolated from the buffy coats of healthy blood donors by Ficoll–Hypaque density gradient (Cytiva, Uppsala, Sweden) centrifugation. Monocytes were then enriched by Percoll density gradient (Cytiva, Uppsala, Sweden) centrifugation achieving 80–90% purity, based on the cell surface expression of CD14, as described previously [58]. Cells were then washed and resuspended in DMEM (Dulbecco’s Modified Eagle Medium; (Corning, Manassas, VA, USA) containing penicillin/streptomycin (1%), glutamine (1%), heat-inactivated fetal bovine serum (FBS) (10%), and heat-inactivated human serum (NHS) (5%) (Complete Medium). Monocytes were seeded into 48-well flat-bottom plastic plates at 2.5 × 10^5^ cells/mL/well and were maintained for an additional 7 days at 37 °C in 5% CO_2_ to promote their full differentiation into MDMs (≥95% CD14+), as described previously [59].

### 4.2. M1 vs. M2 Polarization of MDMs

As we have already extensively published [13,14,24,50], fully differentiated MDMs were stimulated for 18 h with either IFN-γ (20 ng/mL; R&D Systems, Minneapolis, MN, USA) plus TNF-α (2 ng/mL; R&D Systems, Minneapolis, MN, USA) to induce M1 polarization or with IL-4 (20 ng/mL; R&D Systems, Minneapolis, MN, USA) to promote M2 polarization. Subsequently, the culture supernatant containing exogenous cytokines was removed and replaced with complete medium without cytokines before ZIKV infection for the remaining time in culture. Phenotypically, M1-MDMs showed a downregulation of CD4, CD16, CD163, and CD206 vs. CTRL MDMs; conversely, M2-MDMs, which also showed a downregulation of CD4 vs. CTRL MDMs, upregulated CD209 (DC-SIGN) vs. CTRL and M1-MDMs. Other markers, including CD14, CD18, CD80, CD86, and HLA-DR, were not significantly affected by M1/M2 polarization, as published [60].

### 4.3. Zika Virus (ZIKV) Strains and the Plaque Forming Assay (PFA)

The viruses used in this study were either the historical ZIKV strain (MR766), (EVAg—European Virus Archive), or the PRVABC59 strain, obtained from the CDC (GenBank Accession #KU501215). Both viral strains were expanded in Vero cells and their infectious titer was determined by a PFA, as published [61,62]. Briefly, Vero cells (1.2 × 10^6^ cells/well) were seeded in 6-well culture plates. Ten-fold dilutions of virus containing supernatants in culture medium supplemented with 1% heat-inactivated FBS were prepared 24 h later, and 1 mL of each dilution was added to the cells. The plates were incubated for 4 h at 37 °C in 5% CO_2_; then, unadsorbed virus was removed and 2 mL of culture medium supplemented with 1% methylcellulose (Sigma, St. Louis, MO, USA) were added to each well, followed by incubation at 37 °C in 5% CO_2_ for 6 days. The methylcellulose overlay was then removed, and the cells were stained with 1% crystal violet in 70% methanol. Plaques were counted and viral titers were expressed as plaque-forming units per ml (PFU/mL).

### 4.4. ZIKV Infection of Human Monocytes and MDMs

Cells were seeded at a concentration of 2.5 × 10^5^ cells/mL/well and infected at an MOI of 1. Viral supernatants were collected up to 9 days post-infection, and their infectious contents were determined by the PFA. Cells were fixed and the efficiency of infection was also evaluated by immunofluorescence staining with specific anti-ZIKV RNA mAb 3 days post-infection.

### 4.5. Adenylate Kinase (AK) Release Assay of Necrotic Cell Death

Culture supernatants (10 μL) were transferred to a half black 96-well plate (Corning Incorporated, New York, NY, USA). Fifty μL of the AK detection reagent (ToxiLight^®^ BioAssay, Lonza Bioscience, Basel, Switzerland) were added to each well, and the plate was incubated for 10 min at room temperature. Luminescence was measured in a Mithras LB940 Microplate Reader (Berthold Technologies, Bad Wildbad, Germany), and the results were expressed as relative luminescent units (RLUs). This kit quantitatively measures the release of AK activity from damaged cells, thus providing an accurate and sensitive determination of cytolysis [63].

### 4.6. Indirect Immunofluorescence Analysis (IFA) for Double-Stranded RNA (dsRNA)

The efficiency of infection was evaluated by IFA staining. Cells were fixed with 4% paraformaldehyde and, after extensive washing and blocking step in PBS with 10% donkey serum and Triton 100X (0.01% *v*/*v*), were incubated with a mouse anti-dsRNA mAb (Scicons, Newark, CA, USA) for ZIKV detection. Cells were then washed with PBS and incubated for 1 h with anti-mouse Alexa Fluor-488 secondary Ab (ThermoFisher Scientific, Waltham, MA, USA). DAPI (4′,6-diamidino-2-phenylindole) was used to stain the cell nuclei.

### 4.7. Western Blotting (WB) Analysis of MERTK and AXL Receptors

Either monocytes or MDMs were seeded in 6-well culture plates (3 × 10^6^ cells/ well in 3 mL). The expression of MERTK and AXL receptors in human monocytes and MDMs was determined by either WB or Fluorescence-Activated Cell Sorting (FACS) analyses. For WB, the cells were lysed in Nonidet P-40 buffer (50 mM Tris HCl pH 7.5, 150 mM NaCl, 1% Nonidet P-40, and 0.5% wt/vol deoxycholate) containing protease inhibitors. The protein concentration of cell lysates was measured by the Bio-Rad protein assay, based on the Bradford method. Proteins were separated by 10% SDS/PAGE, transferred to a nitrocellulose membrane by electroblotting, and incubated with rabbit anti-MERTK or anti-AXL mAbs (R&D Systems, Minneapolis, MN, USA). Rabbit anti-GAPDH mAb (Cell Signaling Technology, Danvers, MA, USA) served as a normalizer.

### 4.8. FACS Analysis

FACS was performed on uninfected/unpolarized M1- and M2-MDMs. MDMs (0.2 × 10^6^ cells/condition) were detached from plastic adhesion after incubation with Accutase (150 µL/well for 30 min at 37 °C), as described [64]. Cells were spun, and their pellets were resuspended in blocking solution with PBS and 10% FBS for 30′ on ice. Cells were stained with a FITC-conjugated primary mAb (R&D Systems, Minneapolis, MN, USA) for 30′. Flow cytometry for MERTK AXL and TYRO3 expression were performed using a FACS Calibur instrument (Becton Dickinson Italia, Milan, Italy), and the results were analyzed with FlowJo software version 8.4.3 (Tree Star).

### 4.9. Ultrastructural Analysis of MDMs

Both unpolarized and M1-MDMs from three independent donors were seeded in 6-well culture plates (3 × 10^6^ cells/well). MDMs were infected at an MOI of 1 with the PRVABC59 isolate. Three days after infection, the cells were fixed by formaldehyde (4%) and glutaraldehyde (2.5%) in cacodylate buffer and incubated for 5 min at room temperature. The samples were then fixed with osmium tetroxide (2%) in glutaraldehyde (2.5%) in cacodylate buffer for 60 min. Monolayers were dehydrated in graded ethanol, washed in propylene oxide, and infiltrated for 12 h in a 1:1 mixture of propylene oxide and epoxide resin (Epon, Miller-Stephenson Chemical, Sylmar, CA, USA). Cells were then embedded in Epon and polymerized for 24 h at 60 °C. Slices were cut with an ultramicrotome (Ultracut Uct; Leica, Wetzlar, Germany), stained with uranyl acetate and lead citrate, and metalized. The ultrathin sections of either uninfected or infected MDMs were observed through transmission electron microscopy (Hitachi H7000, Hitachi High-Tech Europe GmbH, Krefeld, Germany).

### 4.10. PCR Quantification of ZIKV RNA

Viral RNA was extracted from the supernatant collected after five cycles of freezing and thawing (F/T) of infected macrophages at day 3 post-infection. Next, real-time PCR for the NS5 gene was performed to determine the viral RNA copies present after the F/T cycles. RNA quantification was conducted using the Quanty ZIKA Kit (Clonit, Milan, Italy), which includes a reference curve of viral RNA at known copy numbers.

### 4.11. RNAseq of IFN-Stimulated Genes (ISGs)

*Sample and RNA preparation*. CTRL and M1 cells were obtained from 4 independent donors. Cells were seeded in 48-well plates at 250,000 cells/well. Infection was carried out with the PRVABC59 strain at an MOI of 1. At 18 h post-polarization and at 24 and 72 h post-infection, the culture medium was removed, and 250 µL of TRIzol (Thermo Fisher Scientific, Waltham, MA, USA) were added to each well. For each donor and time point, triplicate samples were prepared for both infected and uninfected cells. To ensure a yield of at least 1 µg of total RNA, the contents of four wells for each condition were pooled. Total RNA was extracted using a Qiagen RNeasy Mini kit following the manufacturer’s instructions (Qiagen, Hilden, Germany).

*RNA Library Preparation and NovaSeq Sequencing*. RNA samples were quantified using a Qubit 4.0 Fluorometer (Life Technologies, Carlsbad, CA, USA), and RNA integrity was checked with an RNA Kit on an Agilent 5600 Fragment Analyzer (Agilent Technologies, Palo Alto, CA, USA). RNA sequencing library preparation was prepared using a NEBNext rRNA Depletion Kit (Human/Mouse/Rat) and NEBNext Ultra II Directional RNA Library Prep Kit for Illumina following the manufacturer’s instructions (NEB, Ipswich, MA, USA). Briefly, rRNA was depleted with a NEBNext rRNA Depletion Kit (Human/Mouse/Rat). rRNA-depleted RNAs were fragmented. First-strand and second-strand cDNA were subsequently synthesized. The second strand of cDNA was marked by incorporating dUTP during the synthesis. cDNA fragments were adenylated at 3′ends, and an indexed adapter was ligated to cDNA fragments. Limited cycle PCR was used for library amplification. The dUTP incorporated into the cDNA of the second strand enabled its specific degradation to maintain strand specificity. Sequencing libraries were validated using a DNA Kit on the Agilent 5600 Fragment Analyzer (Agilent Technologies, Palo Alto, CA, USA) and quantified by using a Qubit 4.0 Fluorometer (Invitrogen, Carlsbad, CA). The sequencing libraries were multiplexed and loaded on the flow cell on the Illumina NovaSeq 6000 instrument according to the manufacturer’s instructions. The samples were sequenced using a 2 × 150 Pair-End (PE) configuration v1.5. Image analysis and base calling were conducted by NovaSeq Control Software v1.7 on the NovaSeq instrument. Raw sequence data (.bcl files) generated from Illumina NovaSeq were converted into fastq files and de-multiplexed using Illumina bcl2fastq program version 2.20. One mismatch was allowed for index sequence identification.

After investigating the quality of the raw data, sequence reads were trimmed to remove possible adapter sequences and nucleotides with poor quality using Trimmomatic v.0.36. The trimmed reads were mapped to the Homo sapiens reference genome (version GRCh38) using the STAR aligner v.2.5.2b [65], and BAM files were generated as a result of this step. Unique gene hit counts were calculated by using feature Counts from the Subread package v.1.5.2 [66], and only unique reads that fell within exon regions were counted.

### 4.12. Statistical Analysis

Statistical analysis was performed using Prism GraphPad software v. 6.0 (GraphPad Software, www.graphpad.com). The results are reported as means ± SDs. Comparison between groups was performed using one-way ANOVA; *t*-test analysis was performed when appropriate, as indicated in the text.

## 5. Conclusions

Primary human monocytes and, in particular, MDMs support productive ZIKV infection in vitro in the absence of clear cytotoxicity. We also provide compelling evidence that the M1, but not M2, polarization of MDMs, obtained by stimulating cells with pro-inflammatory cytokines, curtailed virus replication in these cells. M1 polarization likely restricts ZIKV replication through at least two mechanisms: the downregulation of MERTK, a putative viral entry receptor, from the cell surface and the upregulation of several ISGs. In addition, we report that ZIKV infection leads to the accumulation of cell-associated virions in both unpolarized and M1-MDMs, suggesting the hypothesis that these cells may act as “Trojan horses” of infection, as previously suggested for HIV-1 infection [67]. While our observations highlight a relevant role for M1 polarization of human MDMs in the containment of ZIKV replication, further studies should elucidate the molecular pathways involved in M1 restriction and determine whether “druggable” targets could be identified against ZIKV infection.

## Figures and Tables

**Figure 1 ijms-26-00951-f001:**
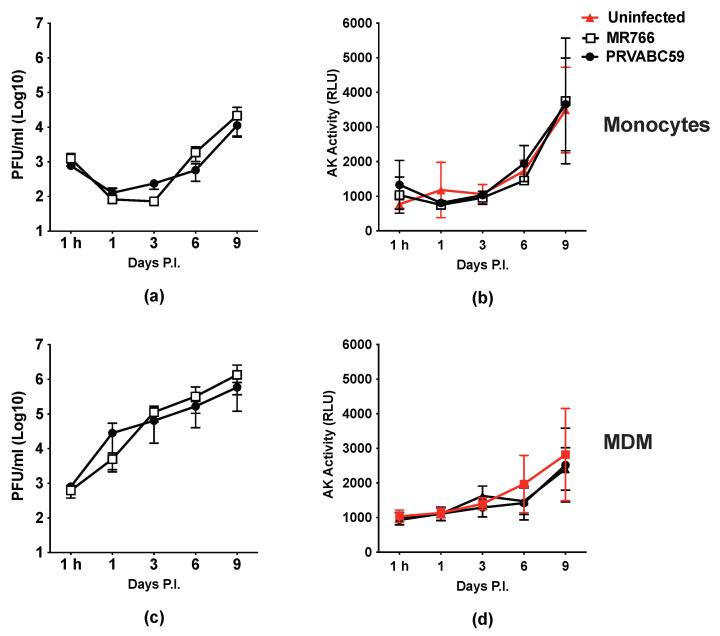
Human monocytes and MDMs are productively infected by ZIKV but resistant to its cytopathic effect. Cells were infected at an MOI of 1 with two ZIKV strains: African 1947 (MR766) and Asian 2015 (PRVABC59). (**a**) Kinetics of viral replication in monocytes measured by the PFA. Data represent the mean ± SD of monocytes obtained from two independent donors, tested in triplicate per time point. (**b**) Necrotic cell death in monocytes measured by AK activity. Data represent the mean ± SD from triplicate at each time point. (**c**) Kinetics of viral replication in MDMs measured by the PFA. Data represent the mean ± SD of MDMs obtained from six independent donors tested in triplicate at each time point. (**d**) Cell death in MDMs measured by AK activity. Data represent the mean ± SD of MDMs obtained from six independent donors, run in duplicates.

**Figure 2 ijms-26-00951-f002:**
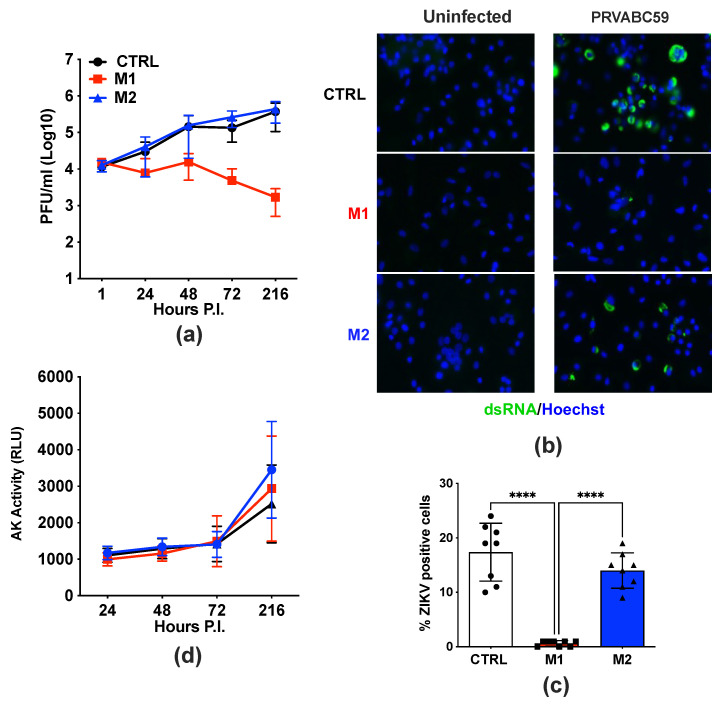
M1-polarized MDMs restrict ZIKV replication compared to CTRL- and M2-MDMs. Cells were infected at an MOI of 1 with ZIKV Asian 2015 strain (PRVABC59). Supernatants were collected up to 9 days post-infection and tested for both infectious viral titers and AK activity. (**a**) Kinetics of viral replication in CTRL, unpolarized, and M1- or M2-MDMs as measured by the PFA, representing the mean ± SD of MDMs obtained from six independent donors, each run in duplicates. (**b**) MDMs were fixed at day 3 post-infection and stained for ZIKV with an anti-dsRNA mAb (ZIKA-green). The blue channel represents DAPI staining for nuclei. Scale bar: 25 µm. Images were obtained using the Zeiss Axio Observer.Z1 microscope (ZEISS AG, Oberkochen, Germany). (**c**) Four different images from MDMs obtained from two independent donors were quantified. Bars represent the mean ± SD of percentage of infected cells. Statistical significance was calculated by one-way ANOVA with the Tukey correction and is indicated by asterisks (**** *p* < 0.0001). (**d**) Cell death measured by AK activity in culture supernatants at different time points representing the mean ± SD of MDMs obtained from six independent donors, each tested in duplicates.

**Figure 3 ijms-26-00951-f003:**
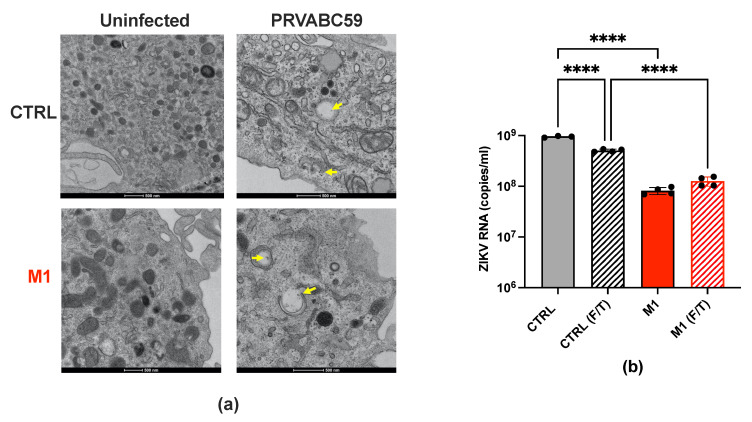
Retention of ZIKV particles in the cytoplasm of CTRL and M1-MDMs. (**a**) TEM images of MDMs obtained from a single donor, representative of three independently tested, showing the presence of virions in the cytoplasm (yellow arrows) and contained in cytoplasmic vacuoles in both unpolarized and M1-polarized MDMs; see also Appendix A. (**b**) PCR-based quantification of extracellular and cell-associated freeze/thaw (F/T) viral RNA. Bar graphs represent the mean ± SD of MDMs from two donors run in duplicates. Statistical significance was calculated by one-way ANOVA with the Tukey correction and it is indicated by asterisks (**** *p* < 0.0001).

**Figure 4 ijms-26-00951-f004:**
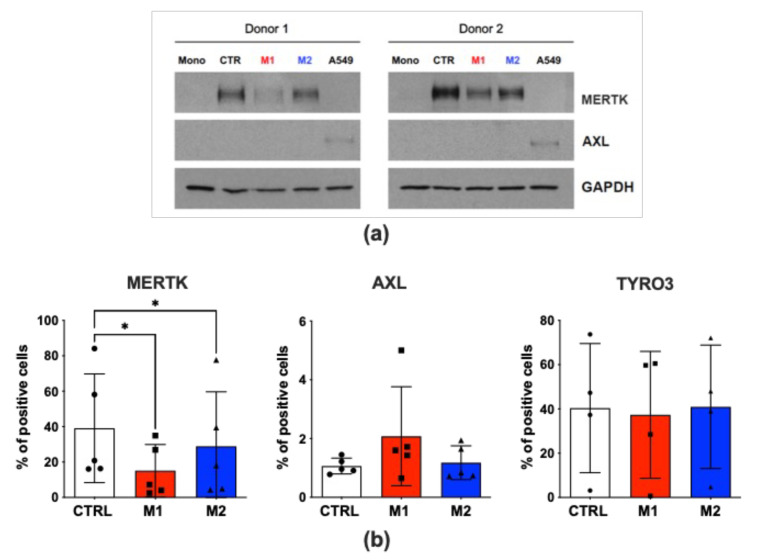
M1 polarization of MDMs downregulates MERTK expression. (**a**) Western blot analysis of putative ZIKV entry receptors in monocytes (Mono) and M1/M2-polarized and control (CTR) MDMs. While AXL was barely detectable, MERTK was clearly expressed in MDMs, although not in monocytes. These results were obtained with cells from two donors, representative of four independently tested. (**b**) FACS analysis of putative ZIKV entry receptors in uninfected control and M1/M2-polarized MDMs. AXL was minimally expressed and not modulated by MDM polarization, whereas both MERTK and TYRO3 were clearly detectable in a substantial fraction of cells. Unlike TYRO3, MERTK was significantly downregulated in M1-MDMs, and also in M2-MDMs although at lower levels, compared to control cells. Bar graphs represent the mean ± SD of MDMs from three donors. Statistical significance was calculated by one-way ANOVA with the Tukey correction and is indicated by asterisks (* *p* < 0.05).

**Figure 5 ijms-26-00951-f005:**
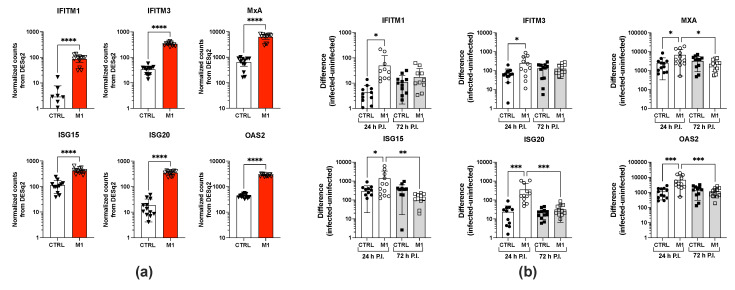
M1 polarization of MDMs upregulates the expression of several ISGs. (**a**) RNAseq was performed to quantify the expression of six ISGs (IFITM1, IFITM3, MXA, ISG15, ISG20, and OAS2) in CTRL, unpolarized, and M1-MDMs at different time points. The kinetic analysis was performed on MDMs obtained from four independent donors and tested in triplicates per time point. (**a**) Expression of ISGs in unstimulated (CTRL) vs. M1-MDMs 18 h after stimulation with polarizing cytokines. Statistical analysis was performed using a *t*-test, and significance levels are indicated as **** *p* < 0.0001. (**b**) ZIKV infection moderately upregulated the expression of ISGs in M1-, but not in CTRL MDMs 24 h after infection, particularly in the cases of ISG20 and OAS2. The results represent the difference in gene expression levels following infection, calculated by subtracting the normalized counts from DSEq2 of uninfected cells from that of infected control cells. CTRL indicates control MDMs, whereas M1 refers to M1-polarized MDMs 24 and 72 h post-infection (PI). Data are derived from RNAseq experiments conducted with four independent donors, each performed in triplicates. The graphs display bars and symbols representing the values of the triplicates from the four donors. Statistical significance, determined by one-way ANOVA with pre-selected pairs of columns, is indicated by asterisks, with significance levels marked as * *p* < 0.05, ** *p* < 0.01, and *** *p* < 0.001.

## Data Availability

The datasets generated and/or analyzed during the current study are available from the corresponding authors on reasonable request.

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
