# Peer review of "Restriction of Zika Virus Replication in Human Monocyte-Derived Macrophages by Pro-Inflammatory (M1) Polarization"

_ijms, 2025, doi:10.3390/ijms26030951_

Round 1
Reviewer 1 Report (New Reviewer)
Comments and Suggestions for Authors
The study by Pagani et al. examined in vitro ZIKV infection in human monocytes and MDMs. The authors demonstrated that MDMs are more permissive to ZIKV and infection does not cause overt CPE. Moreover, M1/M2 polarisation indicates that M1-MDMs can partially restrict ZIKV replication presumably by downregulation of one of its entry receptors and upregulation of OAS1 ISG known for its antiviral functions.
Overall, the manuscript is well-written and provides some insights into ZIKV pathogenicity. The authors have chosen to look at the subset of putative entry receptors and three ISGs to investigate potential mechanisms of ZIKV restriction in M1-MDMs. It seems that a greater emphasis is being placed on downregulation of MERTK as an explanation of the observed effects on ZIKV. However, if all three TAM family receptors are implicated in ZIKV entry, one would assume a redundancy in the system. Why would a moderate downregulation of one of those receptors have a significant effect on viral entry? Moreover, TIM-1 receptor has also been shown to play a crucial role in ZIKV entry and is expressed on monocytes (at least THP-1).
On the other hand, upregulation of ISGs in M1-MDMs appears to be a more robust way to control viral replication. MXA is also upregulated in M1-MDMs (Figure 5c) and has been shown to restrict ZIKV. Is the difference shown in Figure 5 significant? What is the n number for D2 for MXA? It would be of interest to expand the ISG panel and include IFITM3 and ISG20, that have been reported to restrict ZIKV replication. This should not be a problem since the authors already have samples used for other qPCR.
How have the authors confirmed the effectiveness of their M1/M2 polarisation following stimulation? Have they looked at phenotype of these cells (CD64, CD86, CD206) or the expression of NOS2 and ARG1? This should be included in the Materials and Methods.
Minor comments.
Why is AK activity increasing in non-infected cells (both monocytes and MDMs) over 9 hr incubation? The authors indicate that the kit measures AK release from damaged cells.
Figure 2a – what is the control shown on the graph? There is no information either in the text or figure legend. I am assuming these are non-differentiated MDMs?
Order of panels should be fixed to A-D from left to right. It reads A,B,D,C now.
Figure 3 – is panel A showing uninfected MDM controls, which are mentioned in the text above as data not shown?
Check grammar, eg. higher abundance in the formers than in the latter
Author Response
Reviewer n. 1
- “Overall, the manuscript is well-written and provides some insights into ZIKV pathogenicity.”
Reply: we thank the reviewer for the appreciation of our study and its novelty.
- “It seems that a greater emphasis is being placed on downregulation of MERTK as an explanation of the observed effect on ZIKVN. However, if all three TAM family receptors are implicated in ZIKV entry, one would assume a redundancy in the system. Why would a moderate downregulation of one of those receptors has a significant effect on viral entry? Moreover, TIM-1 receptor has also been shown to play a crucial role in ZIKV entry and is expressed on monocytes (at least THP1).”
Reply: We agree with the reviewer regarding the redundancy of the TAM receptor family in mediating ZIKV entry across various cell types, organs and animal models. For example, TAM receptors are not essential for ZIKV infection in mice [1]. However, Axl appears to play a more significant role than other family members in the infection of Neural progenitor cells [2], Sertoli cells [3], astrocytes [4] and microglia [5].
We did not investigate the role of TIM-1 receptor for ZIKV entry, which has been previously described in infected placentas [6]. This decision was based on our inability to observe productive infection of the monocytic THP-1 cell line by ZIKV, contrary to the published report [7]. This discrepancy may be due to differences between the cell lines used.
- “On the other hand, upregulation of ISGs in M1-MDMs appears to be a more robust way to control viral replication…It would be of interest to expand the ISG panel and include IFITM3 and ISG20, that have been reported to restrict ZIKV replication.”
Reply: We have fully embraced the reviewer’ suggestion and revised our experimental approach in order to broaden the range of ISGs potentially involved in ZIKV infection in our cells. Specifically, we conducted RNASeq analysis focusing on ISGs previously reported to play a role in containing ZIKV infection across various cellular models. Our results not only confirm and expand our earlier observations regarding OAS-2 but also identify ISG20 as potential factor in countering viral infection. Notably, both genes, are significantly upregulated in infected vs. uninfected M1-MDM cells 24h after infection, unlike other ISGs. These findings are discussed in detail in the revised manuscript.
- “How have the authors confirmed the effectiveness of their M1/M2polarisation following stimulation? Have they looked at phenotype of these cells (CD64, CD86, CD206) or the expression of NOS2 and ARG1? This should be included in the Materials and Methods.”
Reply: We appreciate the reviewer’s comments and have addressed them accordingly. Our group has been investigating the role and characteristics of M1/M2 polarization in primary human MDM since 2009, primarily in the context of HIV-1 infection, as referenced in the manuscript (refs 13, 14, 23, and 49). In response to the specific reviewer’s request, we have thoroughly characterized the cell surface and intracellular markers of M1/M2 polarization using our protocol in a previous publication [8] in which we observed that CD209/DC-SIGN is uniquely upregulated in M2- MDM and downregulated in M1-MDM among the receptors we examined. To endorse the reviewer’ suggestion, we have added a summary of the markers that distinguish M1 vs. M2 vs. CTRL MDM to the Methods section. Additionally, we have included the specific reference to our previous work in the updated reference list.
Minor comments
- “Why is AK activity increasing in non-infected cells (both monocytes and MDMs) over 9 hr incubation?”
Reply: The observed “high” baseline cellular toxicity is attributable to our protocol, which does not involve changing the culture media after the induction of M1/M2 polarization during the 9-day infection period as MDM are nondividing cells. This approach was consistently applied to both infected and uninfected control cells.
(“Figure 2a – what is the control shown on the graph? There is no information either in the text or figure legend. I am assuming these are non-differentiated MDMs?
Reply: We apologize to the reviewer for the lack of clarity of this figure. We have now clearly indicated the control (CTRL) unpolarized MDM in both the Results section and the legend of Figure 2a.
“Order of panels should be fixed to A-D from left to right. It reads A,B,D,C now.”
Reply: We appreciate the reviewer’s insightful comment. Although we understand the logical point raised, Panel 2c represents the quantification of the images shown in Figure 2b. To enhance clarity, we believe it is more effective to position Panel c directly below Panel b.
- “Figure 3 – is panel A showing uninfected MDM controls, which are mentioned in the text above as data not shown?”
Reply: We confirm that Panel A represents the uninfected control MDM, as mentioned in the text above under “data not shown.” Additionally, we have included a new Supplemental Figure 2 that provides a TEM close-up of several intracytoplasmic vacuoles in uninfected MDM.
- “Check grammar, eg. higher abundance in the formers than in the latter”
Reply: We apologize for any grammatical errors present in the manuscript. We have thoroughly reviewed and corrected these mistakes to ensure clarity and accuracy.
References
- Hastings, A. K.; Yockey, L. J.; Jagger, B. W.; Hwang, J.; Uraki, R.; Gaitsch, H. F.; Parnell, L. A.; Cao, B.; Mysorekar, I. U.; Rothlin, C. V.; Fikrig, E.; Diamond, M. S.; Iwasaki, A., TAM Receptors Are Not Required for Zika Virus Infection in Mice. Cell Rep 2017, 19, (3), 558-568.
- Nowakowski, T. J.; Pollen, A. A.; Di Lullo, E.; Sandoval-Espinosa, C.; Bershteyn, M.; Kriegstein, A. R., Expression Analysis Highlights AXL as a Candidate Zika Virus Entry Receptor in Neural Stem Cells. Cell Stem Cell 2016, 18, (5), 591-6.
- Strange, D. P.; Jiyarom, B.; Pourhabibi Zarandi, N.; Xie, X.; Baker, C.; Sadri-Ardekani, H.; Shi, P. Y.; Verma, S., Axl Promotes Zika Virus Entry and Modulates the Antiviral State of Human Sertoli Cells. mBio 2019, 10, (4).
- Chen, J.; Yang, Y. F.; Yang, Y.; Zou, P.; Chen, J.; He, Y.; Shui, S. L.; Cui, Y. R.; Bai, R.; Liang, Y. J.; Hu, Y.; Jiang, B.; Lu, L.; Zhang, X.; Liu, J.; Xu, J., AXL promotes Zika virus infection in astrocytes by antagonizing type I interferon signalling. Nat Microbiol 2018, 3, (3), 302-309.
- Hastings, A. K.; Hastings, K.; Uraki, R.; Hwang, J.; Gaitsch, H.; Dhaliwal, K.; Williamson, E.; Fikrig, E., Loss of the TAM Receptor Axl Ameliorates Severe Zika Virus Pathogenesis and Reduces Apoptosis in Microglia. iScience 2019, 13, 339-350.
- Nobrega, G. M.; Samogim, A. P.; Parise, P. L.; Venceslau, E. M.; Guida, J. P. S.; Japecanga, R. R.; Amorim, M. R.; Toledo-Teixeira, D. A.; Forato, J.; Consonni, S. R.; Costa, M. L.; Proenca-Modena, J. L.; Zika-Unicamp, N., TAM and TIM receptors mRNA expression in Zika virus infected placentas. Placenta 2020, 101, 204-207.
- Lima, M. C.; Azevedo, E. A. N.; de Morais, C. N. L.; de Sousa, L. I. O.; Carvalho, B. M.; da Silva, I. N.; Franca, R. F. O., The P-MAPA Immunomodulator Partially Prevents Apoptosis Induced by Zika Virus Infection in THP-1 Cells. Curr Pharm Biotechnol 2021, 22, (4), 514-522.
- Cassol, E.; Cassetta, L.; Rizzi, C.; Gabuzda, D.; Alfano, M.; Poli, G., Dendritic cell-specific intercellular adhesion molecule-3 grabbing nonintegrin mediates HIV-1 infection of and transmission by M2a-polarized macrophages in vitro. AIDS 2013, 27, (5), 707-16.
Reviewer 2 Report (New Reviewer)
Comments and Suggestions for Authors
ZIKV, known for its severe neurological effects such as microcephaly in fetuses and Guillain-Barré Syndrome in adults, primarily spreads through mosquitoes but can also be transmitted sexually and from mother to fetus. The virus has the ability to infect human immune cells, including monocytes and monocyte-derived macrophages (MDM), without causing evident cell damage.
In this study, researchers found that while ZIKV replicated in MDM, the virus accumulated in cytoplasmic vacuoles, hinting at unique intracellular dynamics. Importantly, the team explored how cytokine-induced polarization of MDM might influence ZIKV replication. MDM polarized into M1-type cells, which are pro-inflammatory, showed a notable reduction in ZIKV replication. This effect mirrored findings from HIV studies, where M1 polarization also restricted viral growth. In contrast, M2 polarization did not impact ZIKV replication.
The M1 macrophages exhibited reduced expression of MERTK, a receptor that may facilitate ZIKV entry, and increased levels of the antiviral enzyme 2′-5′-oligoadenylate synthetases-2. These findings suggest a dual mechanism—limiting the early steps of the viral life cycle and enhancing antiviral activity—that restricts ZIKV in M1-polarized macrophages. This research points to the promising possibility of antiviral strategies that leverage the body’s innate immune response, offering a new approach in the fight against ZIKV virus.
I would like to commend the authors on a well-structured and insightful manuscript. I have only a small recommendation that could enhance the clarity of the findings presented. I suggest including higher-quality TEM images (Fig.3) of uninfected cells, specifically focusing on the vacuoles and their content, as has been done effectively for infected samples. Furthermore, it would be valuable to include close-up views of these vacuoles to demonstrate the presence of viruses only in infected cells.
Additionally, I am curious about the observations regarding M2-infected cells. Can the author add images TEM on M2 cells infected? Since TEM can be challenging in samples with a low percentage of infected cells, I recommend, if feasible, using immunogold labeling to precisely indicate infected cells. This labeling would greatly strengthen the visual evidence presented and ensure that the focus remains on virus-containing cells.
Author Response
- “I would like to commend the authors on a well-structured and insightful manuscript.”
Reply: We thank the reviewer for his/her deep appreciation of our study!
- “I suggest including higher-quality TEM images (Fig.3) of uninfected cells, specifically focusing on the vacuoles and their content….Furthermore, it would be valuable to include close-up views of these vacuoles to demonstrate the presence of viruses only in infected cells.”
Reply: We have included high-quality TEM images highlighting several intracytoplasmic vacuoles in uninfected MDM as Supplemental Figure 2.
- “I am curious about the observations regarding M2-infected cells. Can the author add images TEM on M2 cells infected? Since TEM can be challenging in samples with a low percentage of infected cells. I recommend, if feasible, using immunogold labeling to precisely indicate infected cells. This labeling would greatly strengthen the visual evidence presented and ensure that the focus remains on virus-containing cells”
Reply: M2-polarized MDM did not exhibit a distinct infection pattern compared to control, unpolarized MDM, as both fully supported ZIKV replication. In contrast, the number of productively infected cells in M1-MDM is too low to be thoroughly examined by TEM, a limitation also noted by the reviewer. We believe that combining TEM images with quantitative analysis of ZIKV RNA provides a clear understanding of the containment of viral replication occurring in M1-MDM.
This manuscript is a resubmission of an earlier submission. The following is a list of the peer review reports and author responses from that submission.
Round 1
Reviewer 1 Report
Comments and Suggestions for Authors
In this paper, Pagani et al have shown that monocyte derived macrophages support productive infection of Zika virus. Polarisation of MDMs to an M2 phenotype does not affect this phenotype, but polarisation to an M1 phenotype abrogates productive replication. Furthermore their preliminary work suggest a decreased expression of zika virus entry receptor, MERTK and increased expression of antiviral restriction factor OAS in M1 macrophages. Overall, the study shows a discrete set of interesting results with some rationale to explain the observed phenomenon. It could be further improved by an independent confirmation of the role of MERTK and OAS2 in zika virus replication. For example, is it possible to downregulate MERTK or block MERTK using an antibody on MDMs to confirm their role in zika virus entry? One study has been published to demonstrate that OAS2 inhibits zika virus replication, but it was done in A549 cells. Would the authors be able to use siRNA to block OAS2 to demonstrate its importance in Zika virus replication?
In addition to these comments please see below some more minor comments:
Figure 1 Y axis label needs to be fixed
How was polarisation to M1 or M2 phenotype confirmed? Surface marker staining (CD80 and DC Sign) needed to confirm polarisation.
Would prefer to see representative histograms of surface entry receptors from figure 4, can be included as a supplementary.
Reviewer 2 Report
Comments and Suggestions for Authors
The authors aim to investigate whether cytokine-induced polarization of MDM into M1 or M2 cells affects ZIKV replication. The study is intriguing; however, many of the experimental designs lack rigor, and some findings are still preliminary. Further clarification and improvement are necessary to address these concerns.
1. The position of panel labels (a, b, …) for each is best placed in the upper left corner.
2. Line 119: the supplemental figure 1 appears to have been lost during the submission process. Please provide the supplemental figure 1 for the revision.
3. Figure 1a, please clearly mark the vertical axis in the chart.
4. In Figure 1, to verify that ZIKV can replicate in the monocytes and MDM cells, it is appropriate to determine progeny virus using a plaque assay. However, given the significance of whether ZIKV can replicate in these cells, the authors should also confirm the kinetic replication using Western Blot analysis (with a specific anti-ZIKV Ab) or qPCR (with specific primers).
5. In Figure 2, please demonstrate that the macrophages are successfully polarized into M1 and M2 cells by measuring the expression of the classical M1 and M2 markers.
6. In Figure 3a, the scale bare is not clear. Please show it clearly or provide the information in the figure legend.
7. In Figure 3a, the intracellular vacuoles may not be viral particles. Please confirm this by staining with an anti-Zika viral protein antibody, an anti-dsRNA antibody, or another appropriate marker.
8. The authors note that M1 polarization induces a downregulation of MERTK and an upregulation of OAS2, which may contribute to the restriction of ZIKV replication in M1 cells. It is not surprising that numerous genes are differentially expressed between unpolarized and M1 macrophages; therefore, it would be valuable to examine the effect of silencing MERTK or overexpressing OAS2 on ZIKV replication in macrophages to make the study more comprehensive.